# The Association between Smoking and Mortality in Women with Breast Cancer: A Real-World Database Analysis

**DOI:** 10.3390/cancers14194565

**Published:** 2022-09-20

**Authors:** Yi-Chen Lai, Yu-Han Chen, Yu-Cih Wu, Fu-Wen Liang, Jhi-Joung Wang, Sher-Wei Lim, Chung-Han Ho

**Affiliations:** 1Department of Emergency Medicine, Tainan Municipal AN-NAN Hospital-China Medical University, Tainan City 70965, Taiwan; 2Department of Family Medicine, Tainan Municipal AN-NAN Hospital-China Medical University, Tainan City 70965, Taiwan; 3Department of Medical Research, Chi-Mei Medical Center, Tainan 71004, Taiwan; 4Department of Public Health, College of Health Sciences, Kaohsiung Medical University, Kaohsiung 80708, Taiwan; 5Department of Medical Research, Kaohsiung Medical University Hospital, Kaohsiung 80756, Taiwan; 6Department of Neurosurgery, Chi Mei Medical Center, Chiali, Tainan 722, Taiwan; 7Department of Nursing, Min-Hwei College of Health Care Management, Tainan 73658, Taiwan; 8Department of Information Management, Southern Taiwan University of Science and Technology, Tainan 71005, Taiwan; 9Cancer Center, Taipei Municipal Wanfang Hospital, Taipei Medical University, Taipei 11696, Taiwan

**Keywords:** breast cancer, smoking, mortality, real-world database

## Abstract

**Simple Summary:**

The association between smoking status and breast cancer mortality in Asian populations has not been extensively studied. In this study, we aimed to investigate the association between smoking status and mortality risk in women diagnosed with breast cancer between 2011 and 2017 using a real-world population database. Women with breast cancer with a history of smoking had a 1.25-fold higher (95% C.I.: 1.08–1.45; *p* = 0.0022) risk of overall mortality and a 1.22-fold higher (95% C.I.: 1.04–1.44; *p* = 0.0168) risk of cancer-specific mortality compared with non-smokers. Smokers who did not have comorbidities showed a significantly higher overall nmortality risk (HR: 1.20; 95% CI: 1.01–1.43; *p* = 0.0408) than non-smokers among women with breast cancer. Additionally, among women with breast cancer who had a history of smoking, current smokers had a 1.57-fold higher risk (95% CI: 1.02–2.42; *p* = 0.0407) of overall mortality compared with ever smokers. It was shown that a current smoking status is significantly associated with an increase in overall and cancer-specific mortality risk in women with breast cancer. Among women diagnosed with breast cancer, those who quit smoking had a lower mortality risk than current smokers. Our results underscore the importance of smoking cessation for women with BC.

**Abstract:**

Smoking increases the cancer-specific and overall mortality risk in women with breast cancer (BC). However, the effect of smoking cessation remains controversial, and detailed research is lacking in Asia. We aimed to investigate the association between smoking status and mortality in women with BC using the population-based cancer registry. The Taiwan Cancer Registry was used to identify women with BC from 2011 to 2017. A total of 54,614 women with BC were enrolled, including 1687 smokers and 52,927 non-smokers. The outcome, mortality, was identified using Taiwan’s cause-of-death database. The association between smoking status and mortality was estimated using Cox proportional regression. Women with BC who smoked had a 1.25-fold higher (95% C.I.: 1.08–1.45; *p* = 0.0022) risk of overall mortality and a 1.22-fold higher (95% C.I.: 1.04–1.44; *p* = 0.0168) risk of cancer-specific mortality compared with non-smokers. The stratified analysis also indicated that women with BC who smoked showed a significantly higher overall mortality risk (HR: 1.20; 95% CI: 1.01–1.43; *p* = 0.0408) than women with BC who did not smoke among women without comorbidities. Additionally, current smokers had a 1.57-fold higher risk (95% CI: 1.02–2.42; *p* = 0.0407) of overall mortality compared with ever smokers among women with BC who smoked. It was shown that a current smoking status is significantly associated with an increase in overall and cancer-specific mortality risk in women with BC. Quitting smoking could reduce one’s mortality risk. Our results underscore the importance of smoking cessation for women with BC.

## 1. Introduction

In 2020, female breast cancer was the most commonly diagnosed cancer and caused 684,996 deaths worldwide [1]. Its incidence rate was also the highest in the female population in Taiwan and far exceeded that of other types of cancer [2]. The prognosis of breast cancer patients is not only dependent on the characteristics of the tumor [3], but it is also closely related to several potentially modifiable lifestyle factors, such as smoking status, alcohol intake, weight control, and physical activity [4,5,6].

Smoking is a well-known all-cause mortality risk in the general population, with causes including vascular diseases, respiratory diseases, and cancers [7,8], and this risk decreases as the number of years after the cessation of smoking increases [9,10]. Smoking has stronger health effects in women than in men regarding the risk of several diseases, such as coronary heart disease, COPD (chronic obstructive pulmonary disease), and specific cancers such as colorectal cancer, bladder cancer, and breast cancer [11,12]. The association between smoking status and the risk of mortality in women with breast cancer has previously been investigated in several studies, but it has not been extensively explored in Asian countries.

Among women with breast cancer, according to the previous literature, current smokers have approximately 1.5 to 3 times higher all-cause mortality rates and 1.2 to 2 times higher cancer-specific mortality rates when compared to never smokers, and the risk of mortality is positively related to the intensity and duration of smoking [13,14,15,16,17]. Several studies also revealed that smoking status was not significantly associated with mortality risk among women with BC, but such literature is relatively rare [18,19]. Tobacco smoke contains hundreds of carcinogenic molecules, and it has been shown to facilitate angiogenesis, tumor growth, and epithelial–mesenchymal transition; it is also associated with axillary lymph node and pulmonary metastasis [20,21,22]. Fortunately, two recent meta-analyses revealed that the cessation of smoking dramatically decreases the risk of cancer-specific mortality to a level near that of never smokers in women with breast cancer, and cessation also reduces the risk of all-cause mortality in these patients [4,23]. However, the literature also revealed that women with breast cancer do not significantly alter their smoking habits after their diagnosis compared to cancer-free women, and the quitting rate was also shown to be lower than that in patients with other cancer types, such as lung cancer and colorectal cancer [4,24,25].

As an important modifiable factor associated with the prognosis of women with breast cancer, smoking habits must be reduced to improve the quality of care and the survival rate of these patients. A detailed characterization of the relationship between smoking and prognosis is vital to encourage smoking cessation when providing education and lifestyle advice to specific groups of women with breast cancer. With this goal in mind, we conducted a population-based, retrospective cohort study to analyze the effect of smoking on mortality risk in women with breast cancer of different ages and with different disease stages, comorbidities, BMIs (body mass indexes), treatment courses, and intensities and durations of smoking habits.

## 2. Materials and Methods

### 2.1. Data Source

The Taiwan Cancer Registry (TCR) was used to identify women with breast cancer. The TCR was established in 1979 to gather information regarding individual demographics, cancer stages, primary cancer sites, tumor histology, and treatment types in patients with cancer to understand the incidence and mortality rates of cancer in Taiwan. Within the registry, the definition of cancer types is based on the International Classification of Diseases for Oncology, third edition (ICD-O-3). The TCR has been used in its short-form and long-form format in different periods to examine different major cancers.

The disease comorbidities of women with breast cancer were collected from the National Health Insurance Research database (NHIRD). The NHIRD contains data from Taiwan’s single-payer population insurance system, in which more than 99% of Taiwan’s 23 million citizens are registered. The diagnosis codes in the NHIRD are based on the International Classification of Diseases, Ninth Revision, Clinical Modification (ICD-9-CM). In addition, the death records are based on the database of death registration for the population of Taiwan.

The above databases are managed by the Health and Welfare Data Science Center (HWDC) of the Ministry of Health and Welfare. Only researchers can apply to use these databases for research purposes. After verification, the HWDC provides the relevant databases and presents de-identified forms to researchers. The HWDC released the above database to us in a de-identified and anonymized format, and we only used the database provided by the HWDC [26]. This study was conducted in compliance with the Declaration of Helsinki of 1964 and has been approved by the Ethics Committee of the Institutional Review Board of Chi-Mei Hospital (IRB:10912-E02). The requirement for informed consent was waived by the Research Ethics Committee of Chi Mei Hospital.

### 2.2. Study Population

Women with new-onset breast cancer (ICD-O-3: C50) from January 2011 to December 2017 were enrolled in this study because TCR started to record the smoking information in 2011. Women with a history of breast cancer before 2011 were excluded. Patients with uncompleted records regarding all of the variables examined were excluded. All of the study subjects were divided into a group of women with breast cancer with a smoking history (current and former smokers) and a group of those without a smoking history (non-smokers). The group with a smoking history included current smokers and former smokers in order to consider the effect of nicotine. According to the guidelines of the Taiwan Cancer Registry, on the date they were initially diagnosed with breast cancer, women were asked by a physician about their smoking status; the presence of smoking, the number of packs smoked per day, and smoking year were determined. Figure 1 illustrates the flowchart of subject selection in the study.

### 2.3. Outcome and Measurements

The primary outcome in this study was mortality, which was identified using Taiwan’s cause-of-death database. Overall mortality and cancer-specific mortality were both considered in this study to avoid bias when attributing the cause of death. 

Considering that potential confounding factors may affect the mortality risk in women diagnosed with breast cancer, age, clinical stage, comorbidities, CCI score, treatment types, behavior, and BMI were measured in this study. In Taiwan, at the age of 45, women can receive free breast cancer screening [27]; therefore, all of the women in the study were classified into four age groups: <45 years, 45–54 years, 55–64 years, and >=65 years. Comorbidities were defined using the ICD-9-CM or ICD-10-CM, including myocardial infarction, congestive heart failure, peripheral vascular disease, cerebrovascular disease, dementia, chronic pulmonary disease, renal disease, hypertension, hyperlipidemia, diabetes, and liver disease (Appendix A). The comorbidities were all presented as yes/no.

Charlson’s comorbidity index (CCI) score is a useful tool to predict mortality risk [28,29], and in this study, we divided the participants’ CCI scores into five groups: 0, 1, 2, 3, and ≥3. The comorbidities and CCI scores were recorded in patients who had at least three outpatient visits or one inpatient visit within one year before the date of the diagnosis of breast cancer to reduce the potential misclassification bias. The measurements regarding health behaviors included drinking alcohol, chewing betel nuts, and BMI. In addition, the treatment types, including operations, radiotherapy, and chemotherapy, were evaluated regarding their relationship with mortality.

### 2.4. Statistical Analysis

The differences in continuous variables and categorical variables between the women with breast cancer who smoked and those who did not smoke were evaluated using Student’s *t*-test and Pearson’s chi-square test, respectively. The Kaplan–Meier method was used to plot the trend of mortality that the follow up for all of the women started on the date of the diagnosis of breast cancer, and the log-rank test was used to compare the risk of mortality between the two groups. Cox proportional regression analysis was used to estimate the association of the risk of mortality with the hazard ratios (HRs) and 95% confidence intervals (CIs). According to the Schoenfeld residuals test, the assessment of proportional hazard assumption was approved. Multivariable Cox regression was used to present a full model which included smoking status, age, clinical stage, drinking, CCI score, BMI, comorbidities, and treatment types. All analyses were conducted using SAS statistical software version 9.4 (SAS Institute, Inc., Cary, NC, USA). The statistical significance was set at a *p*-value < 0.05. The Kaplan–Meier curves were plotted using STATA (version 16; Stata Corp., College Station, TX, USA).

## 3. Results

In this study, 54,614 women with breast cancer were enrolled, including 1687 women who smoked and 52,927 who did not smoke. The baseline characteristics between smokers and non-smokers among women with breast cancer are presented in Table 1. The distributions of age, clinical stages, drinking alcohol, chewing betel nuts, CCI score, and BMI showed significant differences between women with breast cancer who smoked and those who did not.

Figure 2 illustrates the trends regarding overall mortality and cancer-specific mortality development between smokers and non-smokers with breast cancer (log-rank test *p* = 0.1552 for overall and *p* = 0.0473 for cancer-specific mortality).

The overall mortality risk and cancer-specific mortality risk for smokers and non-smokers in women with breast cancer among all of the patients and stratified characteristics are presented in Table 2. After adjusting for age, clinical stage, drinking alcohol, chewing betel nuts, CCI score, BMI, comorbidities, and treatment types, smokers had a 1.25-fold higher (95% C.I.: 1.08–1.45; *p* = 0.0022) risk of overall mortality and a 1.22-fold higher (95% C.I.: 1.04–1.44; *p* = 0.0168) risk of cancer-specific mortality compared with non-smokers. The stratified analysis of different age groups indicated significant overall and cancer-specific mortality risks in women older than 55, and it was shown that smokers had higher mortality risks than non-smokers. It was also shown that women with breast cancer in late clinical stages (III and IV) who also smoked had significant overall mortality risks compared with non-smokers, but there was only borderline significance regarding the cancer-specific mortality risk. In the current study, we also found that smokers had a 1.43-fold higher risk (95% CI: 1.07–1.92; *p* = 0.0164) of overall mortality than non-smokers in women with breast cancer who drank alcohol. Regarding the CCI score, it was shown that smokers with CCI = 0 had significantly higher overall mortality risks (HR: 1.20; 95% CI: 1.01–1.43; *p* = 0.0408) than non-smokers, but the same estimation was not shown regarding cancer-specific mortality. Smokers with CCI > 3 and those that received chemotherapy also showed significantly higher overall and cancer-specific mortality risks than non-smokers.

The comparison between ever smokers and current smokers regarding overall and cancer-specific mortality risks is shown in Table 3. After adjustments of the selected risk factors, current smokers had a 1.57-fold higher risk (95% CI: 1.03–2.44; *p* = 0.0407) of overall mortality compared with ever smokers. Additionally, the stratified analysis of mortality risk between women who quit smoking and those who did not smoke is presented as Table 4. Current smokers who drank alcohol (HR: 3.08; 95% CI: 1.38–6.85; *p* = 0.0058), had a CCI = 0 (HR: 2.01; 95% CI: 1.12–3.61; *p* = 0.0196), and received radiotherapy (HR: 3.05; 95% CI: 1.39–6.69; *p* = 0.0053) showed a significantly higher overall mortality risk compared with former smokers. In terms of cancer-specific mortality risk, the estimated mortality risk was as similar to the overall mortality risk. However, women aged 45–54 (HR: 3.67; 95% CI: 1.18–11.43; *p* = 0.0246) and 55–64 (HR: 10.66; 95% CI: 1.63–69.48; *p* = 0.0134) who currently smoked presented higher risks than women who quit smoking.

## 4. Discussion

This large, real-world database study investigated the association between cigarette smoking and mortality risk in women with breast cancer in Asia. Cigarette smoke contains at least 69 known carcinogens, and some of them have been shown to be capable of reaching human breast tissue [30,31]. *p53* gene mutations and smoking-related DNA adducts found in smokers’ breast tissue explained the positive association between cigarette smoking and the risk of breast cancer [32,33]. Furthermore, the literature also demonstrated that cigarette smoking increases the motility and invasiveness of breast epithelial cells, and smoking is associated with an increased risk of pulmonary metastasis and lymph node metastasis among women with breast cancer [20,22]. In line with previous studies, which were mainly conducted in Western countries, our results revealed that, in women diagnosed with breast cancer, current smokers have a significant increased overall mortality and cancer-specific mortality risk when compared to never smokers, with a mild and more obvious effect on the overall mortality than the cancer-specific mortality risk [4,15,17,23,34]. The only research in Asia that analyzed the association between smoking status and mortality risk in women with breast cancer was conducted in Japan, which included 880 women and revealed that current smokers displayed no significant increase in all-cause death and breast cancer-specific death rates among all of the women [35]; however, their result should be interpreted with caution due to the relatively small sample size. No increases in overall mortality and cancer-specific mortality risks were found in the former smokers in the present study, which implies that better survival outcomes can be achieved after women with breast cancer quit smoking. The majority of the previous literature demonstrated that a former smoking status increases all-cause mortality but not breast-cancer-specific mortality risk [4,15,23]. The effect of a former smoking status on all-cause mortality risk may be associated with the duration and intensity of smoking. A population-based, prospective, observational study conducted by Passarelli et al. revealed that former smokers were only associated with significant increases in all-cause mortality risk when the smoking duration ≧30 years or amount ≧30 pack-years [34].

### 4.1. Effect Modification by Clinical Stage, Treatment Strategy, and Comorbidities

In our study, the impact of smoking cessation on the risk of mortality was shown to be more prominent in women in the later stages of breast cancer. This result is slightly different from previous research. A retrospective cohort study in the US showed that smoking cessation was associated with improved survival status amongst breast cancer survivors across all stages, and the strength of this relationship was stronger in stage 1 (RR = 2.77, *p* = 0.34) than in stage 3 (RR = 1.35, *p* = 0.42) breast cancer [36]. Two prospective cohort studies also revealed a more prominent difference between current and former smokers in patients with local disease than in regional/distant disease [37,38]. The high 5-year survival rate of women with early-stage breast cancer in Taiwan (100% in stage 0 and stage 1 and 94% in stage 2) [39] may mean that the effect of smoking cessation on the survival rate in patients with stage 0–2 cancer appeared to be less significant in our cohort within the study period, leading to differences in the results.

The increases in mortality risk caused by smoking and the significantly lower overall and breast cancer-specific mortality risk found in former smokers than in current smokers were similar among patients receiving various types of breast cancer therapy in this study, which implies that the potential benefits of the cessation of smoking on survival outcomes do not vary with treatment options. It has been well documented that smoking during cancer treatment was associated with the development of secondary malignancy [40] and post-treatment complications [41]. A population-based nested case–control study conducted by Kaufman et al. showed that post-mastectomy radiotherapy (PMRT) sharply increased the risk of second primary lung cancer among ever smokers (adjusted OR = 18.9, 95% CI 7.9–45.4); however, information regarding smoking status was not documented in this study due to serious bias, and as a result, the difference between ex-smokers and current smokers was not compared [42]. A systemic review performed by Wong reported significant differences in multiple outcomes related to adjuvant breast cancer radiation treatment between smokers and non-smokers, including 41.7% of secondary carcinoma, 71.4% of reconstruction outcomes, and 33% of mortality outcomes [43]. Cigarette smoke has been well documented to impair the efficacy of chemotherapy [40,44], affect the liver metabolism of cytotoxic agents, and therefore increase the complication rates of chemotherapy [45]. The literature also revealed that smoking is associated with higher rates of overall and infectious post-operative complications, reconstructive failure, and flap necrosis [46], and these complication rates could be significantly decreased by the cessation of smoking, which was reported in a retrospective cohort study conducted in the US [47]. The results of this study supported these studies and further emphasized the benefit and importance of quitting smoking.

### 4.2. Effect Modification by Age, Alcohol Consumption, and the Duration and Intensity of Smoking

In this study, it was shown that a much higher percentage of smokers drink alcohol compared with non-smokers. A stronger association between smoking and mortality in alcohol consumers was observed in our results. A prospective cohort study conducted in Canada also revealed a more significant increased risk of breast-cancer-specific mortality associated with current smokers in long-term alcohol consumers (HR = 2.91 for moderate/high alcohol consumption; HR = 1.36 for None/Low) [48]. A plausible explanation for this is that the synergistic effects of cigarette smoking and alcohol consumption increase the level of estradiol, which regulates the expression of insulin-like growth factor-I (IGF-I) and stimulates the proliferation of mammary cells [49,50]. The overall and cancer-specific mortality risks for the former smokers were significantly lower than those of the current smokers who were alcohol consumers, which may imply that the cessation of smoking may be strongly associated with an improvement in survival prognosis in this subgroup. This may also explain why our results showed that the death rate for the former smokers was significantly lower than that for the current smokers who were alcohol consumers.

The duration and intensity of cigarette smoking were positively associated with the risk of death in our study. A dose–response relationship was also found in many previous studies. Boone et al. demonstrated that the all-cause mortality risk was significantly higher for women who smoked ≧20 years (HR = 1.47) or ≧35 pack-years (HR = 1.82) in the US [48], Bérubé et al. reported that women who smoked >30 pack-years had significantly higher breast-cancer-specific mortality (HR = 1.52) and all-cause mortality (HR = 1.83) risks in Canada [13], and Kakugawa et al. found that a duration of smoking >21.5 years was associated with significantly higher all-cause (HR = 3.09) and breast cancer-specific death rates (HR = 3.35) among premenopausal women in Japan [35]. The more significant increase in overall mortality and cancer-specific mortality risks related to smoking in patients older than 65 in this study may be related to the possibly longer duration of smoking habits in these patients. In addition, smoking duration and age showed a positive correlation, and this may explain why the effect was less significant among women aged <45 years.

The main strengths of this study include the large sample size contributed by Taiwan’s NHIRD, as well as the strong medical records regarding smoking habits and statuses retrieved from the population cancer registry database, instead of a single questionnaire or brief interview, which carry a substantial risk of underreporting [4,6,35]. As the first population-based study that investigated the association between mortality risk and smoking habits in women with breast cancer in Asia, our study provides important information regarding education and advice in terms of lifestyle adjustment in this patient group. However, there are several limitations to our study. First, the smoking statuses of current, former, and never smokers were determined using the patients’ statuses on the date of their breast cancer diagnosis, but information regarding the specific date of smoking cessation before diagnosis, the time of quitting, and whether smoking was quit after the diagnosis was not obtained from the medical records. This would have had an impact on our interpretation of the results. Second, information regarding passive smoking exposure could not be retrieved from the database, and the effect of passive smoking on breast cancer mortality risk remains controversial according to the previous literature [48,51,52]. Third, information regarding patients’ adherence to breast cancer treatment was not known, as we only obtained binary yes/no information concerning this aspect. Fourth, the HER2/estrogen/progesterone receptor status of the breast cancer patients was not retrieved from the NHIRD database for further adjustment and analysis, and these characteristics may have had confounding effects to the result. Fifth, due to the influence of traditional Taiwanese culture, the smoking rate of Taiwanese women is much lower than that of men [53], which means that the proportion of smokers in the present study was relatively low (2.9%), further limiting the generalizability of our results. However, according to the Taiwan Health and Welfare report of 2020, the smoking rate of adult women over 18 years old in Taiwan was 2.3–4.3% between 2011 and 2017, which was compatible with our result [39]. Last, similar to the issue regarding smoking status, we only measured the confounding factors on the date of breast cancer diagnosis, and these factors may have changed during the period following a patient’s diagnosis, which may have further affected the survival outcomes of the patients.

## 5. Conclusions

We found that ever smokers have a higher risk of overall and breast cancer-specific mortality than never smokers, and former smokers have lower risks of both compared to current smokers. Although the degrees of the benefits on different subgroups vary, there was no disagreement regarding these trends. Our study emphasized the importance of the cessation of smoking in women diagnosed with breast cancer and provided important information for clinicians when giving advice regarding lifestyle adjustment.

## Figures and Tables

**Figure 1 cancers-14-04565-f001:**
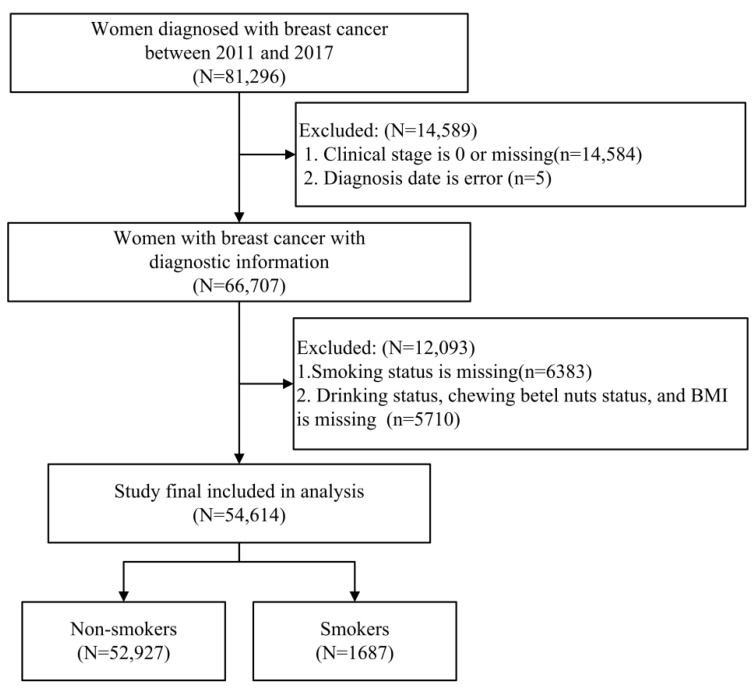
Study selection flowchart.

**Figure 2 cancers-14-04565-f002:**
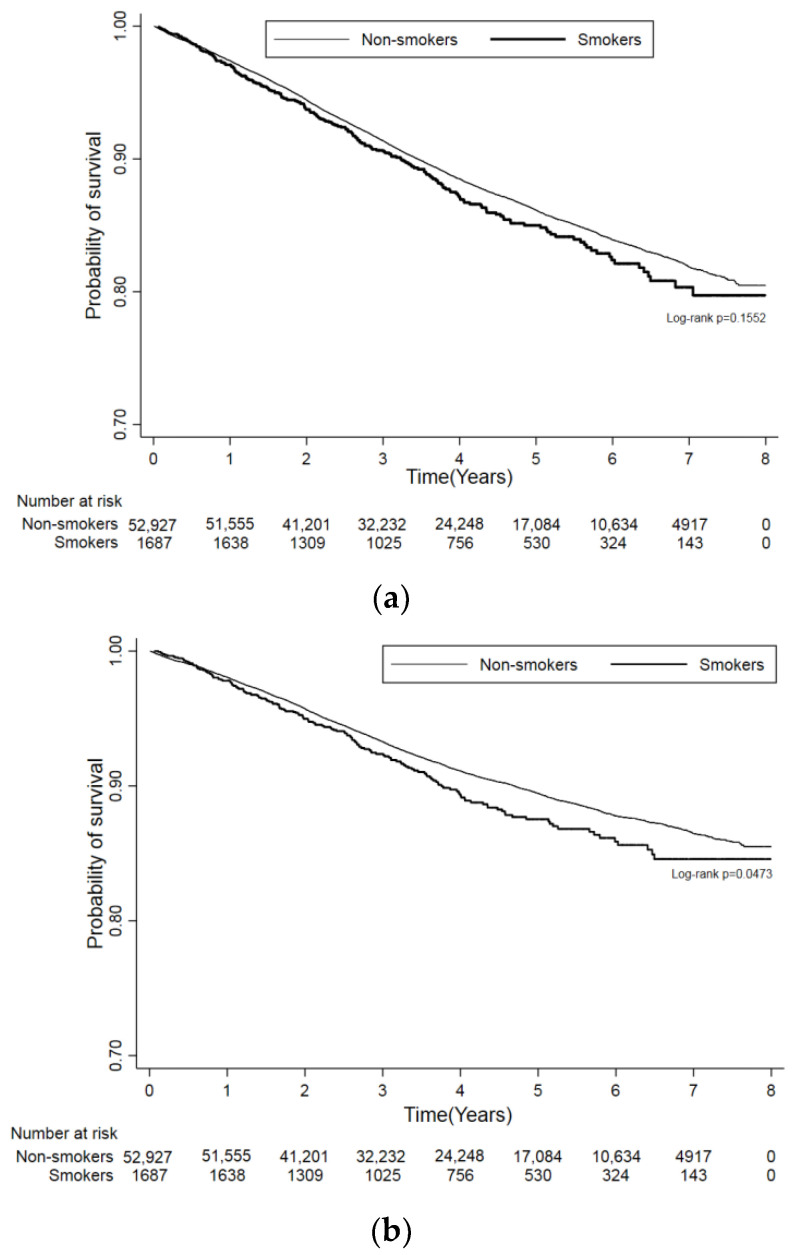
Kaplan–Meier curves for overall mortality (**a**) and cancer-specific mortality (**b**).

**Table 1 cancers-14-04565-t001:** Baseline characteristics between smokers and non-smokers among women with breast cancer.

	Non-Smokers (N = 52,927)	Smokers (N = 1687)	*p*-Value
**Age group, n (%)**			
<45	9681 (18.29)	509 (30.17)	<0.0001
45–54	16,705 (31.56)	676 (40.07)	
55–64	15,118 (28.56)	335 (19.86)	
>=65	11,423 (21.58)	167 (9.90)	
**Clinical Stage, n (%)**			
I	19,578 (36.99)	551 (32.66)	0.0042
II	24,368 (46.04)	827 (49.02)	
III	5195 (9.82)	178 (10.55)	
IV	3786 (7.15)	131 (7.77)	
**Drinking Alcohol, n (%)**	1770 (3.34)	553 (32.78)	<0.0001
**Chewing Betel Nuts, n (%)**	152 (0.29)	90 (5.33)	<0.0001
**CCI, n (%)**			
0	38,854 (73.41)	1317 (78.07)	0.0004
1	7401 (13.98)	203 (12.03)	
2	3628 (6.85)	92 (5.45)	
3	1536 (2.90)	32 (1.90)	
>3	1508 (2.85)	43 (2.55)	
**BMI, n (%)**			
<18.5	2291 (4.33)	130 (7.71)	<0.0001
18.5–25	30,228 (57.11)	1021 (60.52)	
25–30	15,162 (28.65)	384 (22.76)	
30–35	4176 (7.89)	124 (7.35)	
≥35	1070 (2.02)	28 (1.66)	
**Death, n (%)**	6069 (11.47)	211 (12.51)	0.1872
**Death within 5 years, n (%)**	5470 (10.35)	191 (11.32)	0.1905
**Death due to breast cancer, n (%)**	4536 (8.57)	167 (9.90)	0.0554
**Comorbidity, n (%)**			
Myocardial infarction	97 (0.18)	5 (0.30)	0.2488
Congestive heart failure	525 (0.99)	12 (0.71)	0.2502
Peripheral vascular disease	186 (0.35)	8 (0.47)	0.4040
Cerebrovascular disease	1486 (2.81)	36 (2.13)	0.0979
Dementia	476 (0.90)	7 (0.41)	0.0364
Chronic pulmonary disease	1804 (3.41)	78 (4.62)	0.0071
Renal disease	1136 (2.15)	23 (1.36)	0.0280
Hypertension	11,137 (21.04)	252 (14.94)	<0.0001
Hyperlipidemia	8131 (15.36)	184 (10.91)	<0.0001
Diabetes	6068 (11.46)	141 (8.36)	<0.0001
Liver disease	1368 (2.58)	42 (2.49)	0.8085
**Treatment, n (%)**			
Operation	49,499 (93.52)	1574 (93.30)	0.7161
Radiotherapy	29,588 (55.90)	1007 (56.69)	0.0020
Chemotherapy	35,136 (66.39)	1240 (73.50)	<0.0001

**Table 2 cancers-14-04565-t002:** The risk of mortality between smokers and non-smokers among women with breast cancer stratified by characteristics.

		Overall Mortality				Cancer-Specific Mortality			
Smokers vs. Non-Smokers	Patient, N	Death, N (%)	Crude HR (95%CI)	AHR (95% CI) ^⫧^	*p*-Value	Death, N (%)	Crude HR (95%CI)	AHR (95% CI) ^⫧^	*p*-Value
**Overall**	1687	211 (12.51)	1.10 (0.96–1.27)	1.25 (1.08–1.45)	0.0022	167 (9.9)	1.17 (1.00–1.36) *	1.22 (1.04–1.44)	0.0168
**Stratified**									
**Age group**									
<45	509	45 (8.84)	1.16 (0.86–1.57)	1.17 (0.84–1.62)	0.3858	40 (7.86)	1.13 (0.82–1.56)	1.14 (0.81–1.62)	0.4476
45–54	676	70 (10.36)	1.16 (0.91–1.47)	1.19 (0.92–1.54)	0.1851	53 (7.84)	1.00 (0.76–1.32)	1.00 (0.75–1.34)	0.9838
55–64	335	44 (13.13)	1.31 (0.97–1.77)	1.34 (0.98–1.83)	0.0687	38 (11.34)	1.47 (1.06–2.03) *	1.49 (1.06–2.09)	0.0220
>=65	167	52 (31.14)	1.75 (1.33–2.30) *	1.46 (1.10–1.95)	0.0093	36 (21.56)	2.05 (1.47–2.85) **	1.54 (1.09–2.18)	0.0142
**Clinical Stage**									
I	551	17 (3.09)	1.08 (0.67–1.75)	1.56 (0.95–2.57)	0.0791	9 (1.63)	1.45 (0.74–2.81)	1.65 (0.82–3.29)	0.1577
II	827	66 (7.98)	0.92 (0.72–1.17)	1.07 (0.83–1.38)	0.6166	45 (5.44)	0.98 (0.73–1.32)	0.99 (0.73–1.36)	0.9596
III	178	48 (26.97)	1.22 (0.91–1.62)	1.54 (1.12–2.10)	0.0071	39 (21.91)	1.23 (0.89–1.69)	1.40 (0.99–1.99)	0.0554
IV	131	80 (61.07)	1.14 (0.91–1.43)	1.27 (1.00–1.61)	0.0486	74 (56.49)	1.16 (0.92–1.46)	1.26 (0.99–1.62)	0.0630
**Drinking Alcohol**	553	72 (13.02)	1.38 (1.05–1.81) *	1.43 (1.07–1.92)	0.0164	58 (10.49)	1.32 (0.97–1.78)	1.34 (0.97–1.85)	0.0808
**Chewing Betel Nuts**	90	21 (23.33)	1.72 (0.94–3.16)	1.25 (0.56–2.80)	0.5832	17 (18.89)	2.11 (1.04–4.28) *	1.30 (0.33–5.06)	0.7095
**CCI**									
0	1317	144 (10.93)	1.18 (1.00–1.40) *	1.20 (1.01–1.43)	0.0408	118 (8.96)	1.14 (0.95–1.38)	1.11 (0.91–1.35)	0.3052
1	203	19 (9.36)	0.78 (0.49–1.22)	0.99 (0.61–1.59)	0.9626	17 (8.37)	0.98 (0.61–1.59)	1.13 (0.68–1.89)	0.6370
2	92	17 (18.48)	1.11 (0.68–1.79)	1.56 (0.94–2.69)	0.0843	14 (15.22)	1.54 (0.90–2.63)	2.18 (1.23–3.89)	0.0081
3	32	9 (28.13)	1.39 (0.72–2.70)	1.28 (0.63–2.62)	0.4947	5 (15.63)	1.41 (0.58–3.43)	0.77 (0.29–2.09)	0.6117
>3	43	22 (51.16)	1.38 (0.90–2.11)	2.02 (1.28–3.19)	0.0025	13 (30.23)	1.74 (1.00–3.03)	2.69 (1.48–4.89)	0.0012
**Treatment**									
Operation	1574	141 (8.96)	1.10 (0.93–1.30)	1.19 (1.00–1.42)	0.0558	106 (6.73)	1.19 (0.98–1.44)	1.12 (0.92–1.38)	0.2627
Radiotherapy	1007	92 (9.14)	1.12 (0.91–1.38)	1.28 (1.03–1.59)	0.0286	71 (7.05)	1.07 (0.85–1.36)	1.18 (0.92–1.52)	0.1857
Chemotherapy	1240	173 (13.95)	1.21 (1.04–1.41) *	1.35 (1.15–1.59)	0.0003	143 (11.53)	1.19 (1.01–1.41) *	1.28 (1.07–1.53)	0.0063

* *p* < 0.05, ** *p* < 0.0001. **^⫧^** Adjusted for age groups (<45, 45–54, 55–64, and >=65), clinical stage (I, II, III, and IV), drinking alcohol, chewing betel nuts, CCI groups (0, 1, 2, 3, and >3), BMI groups (<18.5, 18.5–25, 25–30, 30–35, and ≥35), comorbidities (myocardial infarction, congestive heart failure, peripheral vascular disease, cerebrovascular disease, dementia, chronic pulmonary disease, renal disease, hypertension, hyperlipidemia, diabetes mellitus, and liver disease), and treatments (operation, radiotherapy, and chemotherapy).

**Table 3 cancers-14-04565-t003:** The risk of mortality between ever smokers and current smokers.

		Overall Mortality					Cancer-Specific Mortality				
	Patient, N	Death, N (%)	Crude HR (95% CI)	*p*-Value	AHR (95% CI) ^⫧^	*p*-Value	Death, N (%)	Crude HR (95% CI)	*p*-Value	AHR (95% CI) ^⫧^	*p*-Value
**Quit smoking**											
Yes	209	27 (12.92)	Ref.		Ref.		22 (10.53)	Ref.		Ref.	
No	1478	184 (12.45)	0.98 (0.65–1.47)	0.9147	1.57 (1.02–2.42)	0.0407	145 (9.81)	0.95 (0.61–1.48)	0.8110	1.48 (0.91–2.40)	0.1113
**Smoking count**											
0.5 pack/day	828	95 (11.47)	Ref.		Ref.		76 (9.18)	Ref.		Ref.	
1 pack/day	734	95 (12.94)	1.17 (0.88–1.55)	0.2931	1.08 (0.79–1.46)	0.6419	76 (10.35)	1.16 (0.85–1.60)	0.3488	1.09 (0.77–1.54)	0.6427
>1 pack/day	125	21 (16.8)	1.46 (0.91–2.34)	0.1158	1.29 (0.78–2.14)	0.3253	15 (12.00)	1.31 (0.75–2.28)	0.3425	1.15 (0.64–2.09)	0.6401
**Smoking year**											
0–10	568	52 (9.15)	Ref.		Ref.		44 (7.75)	Ref.		Ref.	
11–20	587	65 (11.07)	1.25 (0.87–1.80)	0.2323	1.05 (0.71–1.55)	0.8024	49 (8.35)	1.11 (0.74–1.67)	0.6079	1.00 (0.65–1.55)	0.9905
21–30	383	58 (15.14)	1.79 (1.23–2.60)	0.0023	1.09 (0.72–1.64)	0.7007	44 (11.49)	1.60 (1.05–2.43)	0.0284	1.00 (0.62–1.60)	0.9896
>30	149	36 (24.16)	2.88 (1.88–4.40)	<0.0001	1.48 (0.87–2.49)	0.1458	30 (20.13)	2.82 (1.78–4.49)	<0.0001	1.84 (1.02–3.31)	0.0426
**Age group**											
<45	509	45 (8.84)	Ref.		Ref.		40 (7.86)	Ref.		Ref.	
45–54	676	70 (10.36)	1.24 (0.85–1.80)	0.2670	1.35 (0.90–2.02)	0.1483	53 (7.84)	1.05 (0.70–1.58)	0.8224	1.12 (0.71–1.75)	0.6238
55–64	335	44 (13.13)	1.67 (1.10–2.54)	0.0153	1.33 (0.82–2.15)	0.2418	38 (11.34)	1.62 (1.04–2.53)	0.0335	1.14 (0.67–1.94)	0.6193
>=65	167	52 (31.14)	4.37 (2.93–6.52)	<0.0001	2.28 (1.34–3.88)	0.0024	36 (21.56)	3.38 (2.15–5.31)	<0.0001	1.52 (0.83–2.79)	0.1773
**Clinical stage**											
I	551	17 (3.09)	Ref.		Ref.		9 (1.63)	Ref.		Ref.	
II	827	66 (7.98)	2.48 (1.45–4.22)	0.0009	2.15 (1.24–3.72)	0.0064	45 (5.44)	3.21 (1.57–6.56)	0.0014	2.75 (1.32–5.70)	0.0067
III	178	48 (26.97)	10.59 (6.09–18.41)	<0.0001	9.64 (5.34–17.42)	<0.0001	39 (21.91)	16.28 (7.89–33.62)	<0.0001	14.81 (6.94–31.64)	<0.0001
IV	131	80 (61.07)	36.95 (21.82–62.54)	<0.0001	15.79 (8.37–29.81)	<0.0001	74 (56.49)	65.09 (32.50–130.35)	<0.0001	28.73 (12.92–63.91)	<0.0001
**Drinking alcohol** **,** **yes vs. no**	553	72 (13.02)	1.01 (0.76–1.35)	0.9311	1.13 (0.82–1.55)	0.4548	58 (10.49)	1.04 (0.76–1.43)	0.8101	1.22 (0.86–1.74)	0.2719
**Chewing betel nuts** **,** **yes vs. no**	90	21 (23.33)	2.01 (1.28–3.15)	0.0024	0.92 (0.55–1.54)	0.7558	17 (18.89)	2.05 (1.24–3.38)	0.0051	0.80 (0.45–1.43)	0.4467
**CCI**											
0	1317	144 (10.93)	Ref.		Ref.		118 (8.96)	Ref.		Ref.	
1	203	19 (9.36)	N/A	N/A	0.97 (0.54–1.75)	0.9238	17 (8.37)	0.90 (0.54–1.50)	0.6987	1.28 (0.66–2.47)	0.4696
2	92	17 (18.48)	0.83 (0.51–1.34)	0.4375	2.22 (1.10–4.49)	0.0260	14 (15.22)	1.77 (1.02–3.08)	0.0431	2.84 (1.28–6.32)	0.0103
3	32	9 (28.13)	1.77 (1.07–2.93)	0.0261	1.61 (0.61–4.27)	0.3406	5 (15.63)	1.88 (0.77–4.59)	0.1683	1.35 (0.37–4.87)	0.6496
>3	43	22 (51.16)	2.82 (1.44–5.54)	0.0026	6.79 (3.25–14.21)	<0.0001	13 (30.23)	4.78 (2.70–8.49)	<0.0001	6.55 (2.73–15.73)	<0.0001

**^⫧^** Adjusted for smoking count (0.5, 1, and >1 pack/day), smoking year (0–10, 11–20, 20–30, and >30), age groups (<45, 45–54, 55–64, and >=65), clinical stage (I, II, III, and IV), drinking alcohol, chewing betel nuts, CCI groups (0, 1, 2, 3, and >3), BMI groups (<18.5, 18.5–25, 25–30, 30–35, and ≥35), comorbidities (myocardial infarction, congestive heart failure, peripheral vascular disease, cerebrovascular disease, dementia, chronic pulmonary disease, renal disease, hypertension, hyperlipidemia, diabetes mellitus, and liver disease), and treatments (operation, radiotherapy, and chemotherapy).

**Table 4 cancers-14-04565-t004:** The stratified analysis of mortality risk between women who quit smoking and those who did not.

	Overall Mortality		Cancer-Specific Mortality	
Quit smoking, no vs. yes	AHR ^⫧^ (95%CI)	*p*-Value	AHR ^⫧^ (95%CI)	*p*-Value
**Stratified**				
**Age group**				
<45	1.86 (0.61–5.68)	0.2788	1.75 (0.58–5.32)	0.3249
45–54	2.24 (0.90–5.59)	0.0850	3.67 (1.18–11.43)	0.0246
55–64	4.10 (1.00–16.85)	0.0508	10.66 (1.63–69.48)	0.0134
>=65	1.07 (0.42–2.72)	0.8831	0.72 (0.24–2.13)	0.5530
**Clinical Stage**				
Early stage	1.33 (0.63–2.83)	0.4537	1.76 (0.61–5.07)	0.2927
Last stage	1.74 (0.97–3.12)	0.0625	1.51 (0.83–2.75)	0.1769
**Drinking Alcohol**	3.08 (1.38–6.85)	0.0058	3.93 (1.51–10.18)	0.0049
**CCI**				
0	2.01 (1.12–3.61)	0.0196	2.21 (1.16–4.20)	0.0157
1–2	0.99 (0.24–4.13)	0.9876	1.21 (0.24–6.06)	0.8174
>=3	0.38 (0.06–2.57)	0.3240	N/A	0.9983
**Treatment**				
Operation	1.67 (0.91–3.04)	0.0971	1.75 (0.85–3.62)	0.1281
Radiotherapy	3.05 (1.39–6.69)	0.0053	2.89 (1.19–7.02)	0.0193
Chemotherapy	1.53 (0.93–2.54)	0.0953	1.66 (0.96–2.88)	0.0726

**^⫧^** Adjusted for smoking count (0.5, 1, and >1 pack/day), smoking year (0–10, 11–20, 20–30, and >30), age groups (<45, 45–54, 55–64, and >=65), clinical stage (I, II, III, and IV), drinking alcohol, chewing betel nuts, CCI groups (0, 1, 2, 3, and >3), BMI groups (<18.5, 18.5–25, 25–30, 30–35, and ≥35), comorbidities (myocardial infarction, congestive heart failure, peripheral vascular disease, cerebrovascular disease, dementia, chronic pulmonary disease, renal disease, hypertension, hyperlipidemia, diabetes mellitus, and liver disease), and treatments (operation, radiotherapy, and chemotherapy).

## Data Availability

The data sources are the Taiwan Nation Health Insurance Database and Taiwan Cancer Registry. The data are available with permission from the Taiwan Health and Welfare Data Science Center (https://dep.mohw.gov.tw/DOS/np-2497-113.html, accessed on 19 June 2022). Restrictions apply to the availability of these data, which were used under license for this study.

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
