# Peer review of "The Association between Smoking and Mortality in Women with Breast Cancer: A Real-World Database Analysis"

_cancers, 2022, doi:10.3390/cancers14194565_

Round 1
Reviewer 1 Report
The current manuscript describes an impressive database of >57000 Taiwanese women with breast cancer and known smoking status. As suggested by the US Surgeon General, the study attempts to clarify the role of smoking cessation on the subsequent cancer outcome, while describing the role of smoking on breast cancer survival on Asian population, a research field also in need. In this aspect, the paper has some strengths.
However, there are several methodological problems in the study setting, as listed below, and I suggest that the current manuscript would be suitable to an another journal. The English is good in most parts of the text, but some sentences are misleading and poorly worded.
1) The biggest drawback is the binary classification of smoking status, where former smokers are apparently classified along never smokers. This causes significant bias to the reference group and makes conclusions unreliable.
2) Definition of smoking status is also poorly described in 2.2 lines 113. Who determined the smoking status (doctor / questionnaire)? How was the data structured? What timeframe was considered to result in non-smoker?
3) The main finding of the study is that smoking cessation decreased overall mortality (table 3), and a favourable trend in breast cancer mortality, but number of patients was only ~1700, severely limiting the generalization of the results, in view of the current journal's impact.
4) Discussion 4.1 line 230. This is actually not shown in the text, with respect to comment below, table 3 uses multiple reference values, showing that survival does worsen as stage increases, but this has nothing to do with the magnitude of smoking's adverse effect on survival according to stage. Subgroup analysis is suggested. Discussion is also too lenghty.
5) Statistical approach is poorly described, does statistical "adjusting" mean multivariable testing? What are "references" in the tables.
6) Abstract lines 43 and 46 are redundant. See comment #3
7) Ethics. It is unclear which databases are public, and how can they be combined if they are anonymized
Minor remarks
-text should refer to women with brca instead of patients with brca, since the data includes women only
-Some studies show no difference in post-diagnosis survival among women with breast cancer according to smoking status, and should be included in introduction
-DCIS patients (stage 0) should be excluded
-Move the extensive icd code list to supplementary or use Charslon indexing reference
-decreasing cancer risk after smoking cessation is proven in lung cancer, and cannot be generalized (intro line 58)
-study lacks her2/estrogen receptor data, which should be acknowledged
Reviewer 2 Report
Overall Comments
This is a report of an observational study of the association between smoking, smoking cessation, and mortality for women who developed breast cancer in Taiwan between 2011 and 2017. The data sources are appropriate for the research question. More details are needed to clarify the statistical method. The authors could also clarify what is novel about the paper. Below, I have provided some suggestions and questions to improve each section of the manuscript.
A. Simple Summary
Line 22 – the “effect of smoking cessation” generally is not controversial. Specifically, you mean that the impact of smoking cessation on breast cancer mortality is not well studied. Also, this paper is not solely focused on smoking cessation, but rather smoking history generally, so you might revise the first sentences in the Simple Summary and the Abstract.
Line 24 and on – “breast cancer patients” – might be more accurate to say “women diagnosed with breast cancer between 2011 and 2017.”
Line 24 and on – “with smoking”. The usual language for this is “women with a history of smoking” or something similar.
Line 31 – “Quitting smoking…” This sentence should be adjusted so that it is supported by the study. A more accurate summary would be: “among women diagnosed with breast cancer, those who quit smoking (either before or after diagnosis?) had lower mortality than current smokers”.
B. Abstract
Line 36 – “Real-world database” is not informative. Perhaps change throughout to “population-based cancer registry”.
Line 42 – “Smokers without comorbidities…”. More accurate to say that “the association between smoking and hazard of overall mortality was greater for women with comorbidities compared to women without comorbidities”.
C. Introduction
Line 57 – minor note: suggest removing informal language like, “is notorious for”.
Line 73 – ceasing smoking decreases mortality, so need to explain what is new in this study. Is it overall mortality?
D. Methods
Line 111. Clarify that women were included if they received a diagnosis of breast cancer between 2011 and 2017. Also, justify why these years only.
Figure – justify excluding those with missing clinical stage.
– Nearly 10% of patients were missing smoking status – did you investigate whether their
mortality differed from those that had smoking status?
Line 122 – confounders are factors that are causally associated with the outcome (mortality) and associated with the predictor (smoking). Not sure all of these factors meet this criterion, but may still be interesting for effect modification. Need a better justification for which variables were included in the analysis.
Line 126 – ICD codes could be moved to a supplementary table.
Line 145 – there needs to be a better justification for restricting comorbidities and CCI to patients with at least 3 outpatient or 1 inpatient visit prior to the diagnosis date.
Line 154 – clarify whether follow-up started at the diagnosis date.
You should clarify how the “comorbidities” covariate is parameterized. Is it a yes/no?
E. Results
Line 163 – 1,780 / (1,780 + 56,000) = 2.9% ever-smoking prevalence seems low. You might comment on whether this is consistent with the population of Taiwan or other studies of women diagnosed with breast cancer.
In Table 1, the statistical significance is not so useful for such a large study. You might comment that the smokers tended to be younger (surprising to me); that there was essentially no difference in clinical stage at time of diagnosis; that smokers had less comorbidities (also surprising); that BMI was similar. These observations could help justify inclusion of these covariates in the model.
What is the absolute 5-year risk of mortality for smokers vs. non-smokers.
Line 176 – comorbidities, such as CVD, may be in the path between treatment (chemotherapy) and outcome (mortality). Something to consider. Were risks proportional in the Proportional Hazards model?
Table 2.
Stratified models –
Does smoking duration differ by age? Could this explain why the effect is smaller for women aged <45 years at time of diagnosis?
The strong effect for clinic stage 0 is based on a small number of deaths. You might comment on this.
Table 3. Why does adjustment bring the “Smoking Quit” Crude HR from 0.99 to an AHR of 1.57?
Why did you not consider pack-years, which is the standard measure?
F. Discussion
Line 220-221 – “this study” refers to the present study?
Line 230 – I think that the differences in smoking effects across stages deserves more discussion. It is plausible that the smoking effect would be more pronounced among women with lower stage that have a longer survival time and are less likely to die of their cancer overall (therefore more likely to see smoking-specific effects), but this is not the conclusion in the study.
Line 261 – There is a lot of discussion about the interaction between smoking and treatment, but I don’t see any analysis stratified by treatment or interaction analysis in this study.
Line 265 – There is no formal test of interaction between smoking and drinking (or any effect modifiers).
Line 299 – It should be more clear at the beginning of the paper that you don’t have the time of quitting, and that it may not have been after diagnosis, despite mentions of smoking cessation after diagnosis in the Introduction.
Conclusions.
Consider being more cautious in conclusion. Former smokers had a lower risk of mortality than current smokers. Ever-smokers had a higher risk of mortality than never-smokers.
Reviewer 3 Report
The submitted study is the the first population-based report on the association between breast cancer (BC) mortality nad smoking habits in Asian women. Based on data from over 57000 patients, the study confirms that the duration and intensity of tobacco smoking negativly impact the risk of death from BC. The impact of smoking cessation on mortality risk was more pronounced in the later stages of BC. The topic is within the scope of the journal. The high quality of the data obtained predestines the manuscript for publication. The reliability of the data is additionally increased by a relatively short time window (2011-2017), so that the possible effect of BC treatment variability could be minimized. The only flaws in the submitted manuscript I could identify in the discussion. The authors should briefly recap the basic data explaining a possible mechanism between tobacco smoking and breast cancer incidence and mortality. This aspect is addressed in just one sentence when it comes to the modulating effect of heavy alcohol consumption. In addition, the authors should address the possibility of other potential confounding factors that were not included in the evaluation. Finally, I missed the disussion with two earlier studies dealing with the same topic in non-Asian women: Wartenberg et al. Passive smoking exposure and female breast cancer mortality. J Natl Cancer Inst. 2000;92:1666-73, as well as Parada et al. Environmental Tobacco Smoke Exposure and Survival Following Breast Cancer. Cancer Epidemiol Biomarkers Prev. 2017;26:278-280. Overall a very useful study with only a few minor concerns.Author Response
Please see the attachment.

Reviewer 4 Report
Paper Title: The association between smoking and mortality in patients with breast cancer: a real-world database analysis
Despite the fact that this topic is interesting, and I really think that the authors have devoted a lot of effort to their research, this current manuscript still needs some revisions.
1. There is a very large number of typographical errors throughout the manuscript.
2. The abbreviations in the tables should be interpreted.
3. For age, the authors used 10 years as the age classification, why did the author not distinguish between people under 45 years older? It may provide special results for different people in different age ranges. Please clarify.
Round 2
Reviewer 1 Report
The authors have done excellent job by revising the original paper. The English language is well-written. I have only a minor remark. Term "non-smoker" should refer to "never smoker" as now explained in the author responses and the text. This is especially important in the abstract and summary, to avoid misunderstanding.
Reviewer 2 Report
The authors have carefully and thoroughly addressed my concerns.